

# Active management is required to turn the tide for depleted *Ostrea edulis* stocks from the effects of overfishing, disease and invasive species

Luke Helmer[1], Paul Farrell[1], Ian Hendy[1,2], Simon Harding[2,3], Morven Robertson[2] and Joanne Preston[1]

[1] Institute of Marine Sciences, University of Portsmouth, Portsmouth, Hampshire, UK
[2] Blue Marine Foundation, London, UK
[3] Current affiliation: Institute of Marine Resources, The University of the South Pacific, Suva, Fiji

Corresponding author
Joanne Preston,
joanne.preston@port.ac.uk

## ABSTRACT

The decline of the European oyster *Ostrea edulis* across its biogeographic range has been driven largely by over-fishing and anthropogenic habitat destruction, often to the point of functional extinction. However, other negatively interacting factors attributing to this catastrophic decline include disease, invasive species and pollution. In addition, a relatively complex life history characterized by sporadic spawning renders *O. edulis* biologically vulnerable to overexploitation. As a viviparous species, successful reproduction in *O. edulis* populations is density dependent to a greater degree than broadcast spawning oviparous species such as the Pacific oyster *Crassostrea* (*Magallana*) *gigas*. Here, we report on the benthic assemblage of *O. edulis* and the invasive gastropod *Crepidula fornicata* across three actively managed South coast harbors in one of the few remaining *O. edulis* fisheries in the UK. Long-term data reveals that numbers of *O. edulis* sampled within Chichester Harbour have decreased by 96%, in contrast numbers of *C. fornicata* sampled have increased by 441% over a 19-year period. The recent survey data also recorded extremely low densities of *O. edulis,* and extremely high densities of *C. fornicata*, within Portsmouth and Langstone Harbours. The native oyster's failure to recover, despite fishery closures, suggests competitive exclusion by *C. fornicata* is preventing recovery of *O. edulis*, which is thought to be due to a lack of habitat heterogeneity or suitable settlement substrate. Large scale population data reveals that mean *O. edulis* shell length and width has decreased significantly across all years and site groups from 2015 to 2017, with a narrowing demographic structure. An absence of juveniles and lack of multiple cohorts in the remaining population suggests that the limited fishing effort exceeds biological output and recruitment is poor. In the Langstone & Chichester 2017 sample 98% of the population is assigned to a single cohort (modal mean 71.20 ± 8.78 mm, maximum length). There is evidence of small scale (<5 km) geographic population structure between connected harbors; the 2015 Portsmouth and Chichester fishery populations exhibited disparity in the most frequent size class with 36% within 81–90 mm and 33.86% within 61–70 mm, respectively, the data also indicates a narrowing demographic over a short period of time. The prevalence of the disease Bonamiosis was monitored and supports this microgeographic population structure. Infection rates of *O. edulis* by

*Bonamia ostreae* was 0% in Portsmouth Harbor (*n* = 48), 4.1% in Langstone (*n* = 145) and 21.3% in Chichester (*n* = 48) populations. These data collectively indicate that *O. edulis* is on the brink of an ecological collapse within the Solent harbors. Without effective intervention to mitigate the benthic dominance by *C. fornicata* in the form of biologically relevant fishery policy and the management of suitable recruitment substrate these native oyster populations could be lost.

## INTRODUCTION

The habitat of the European flat oyster *Ostrea edulis* (Linnaeus, 1758) includes a range of firm substrata from the lower intertidal to subtidal depths up to 80 m (*Perry & Jackson, 2017*) across a biogeographic range that stretches from Morocco, throughout the Mediterranean and Black seas, to Norway (*Lallias et al., 2010*). Earliest records identify *O. edulis* shell middens from the Mesolithic period, (*Gutiérrez-Zugasti et al., 2011*) and cultivation from the Roman Empire (*Gunther, 1897*), illustrating the long history of extraction for human consumption.

Native oyster populations throughout Britain remained large and lightly fished up until the early 19th century (*Edwards, 1997*, *Key & Davidson, 1981*). By the mid-19th C demand was high, approximately 700 million oysters were consumed in London during 1864, supporting a sizable UK fleet of 120,000 oyster dredgers (*Philpots, 1890* in *Edwards, 1997*). In France, historic shell piles contained approximately $5 \times 10^{12}$ oysters (*Gruet & Prigent, 1986* in *Goulletquer & Heral, 1997*) highlighting their vast densities and the likely unsustainable extraction of this species. Despite UK governmental legislation put in place by a parliamentary committee (still enforced under the Sea Fisheries (Shellfish) Act of 1967), stocks inevitably declined. Landings of *O. edulis* in English and Welsh waters decreased from 3,500 tonnes in 1887, to <500 tonnes in 1947 (*Laing, Walker & Areal, 2006*). The distribution of *O. edulis* across the UK and Europe is a fraction of reported historic levels, with many historic beds completely depleted and the few remaining populations found predominantly in subtidal habitats (*Gross & Smyth, 1946*; *Laing, Walker & Areal, 2005*, *2006*; *Culloty & Mulcahy, 2007*). This is a global issue with approximately 85% of the world's oyster populations and their associated habitat having been lost (*Beck et al., 2011*), resulting in ecological decline due to the ecosystem services oysters provide (*Cranfield, Michael & Doonan, 1999*, *Carbines, Jiang & Beentjes, 2004*).

This decline in the native oyster is reflected in the Solent, described in detail by *Key & Davidson (1981)*. Oyster cultivation within the Solent occurred from 1866 onward, with small scale removal and relocation taking place in the early 20th century (*Anon, 1912–1940* in *Key & Davidson (1981)*). Throughout the 1970s and early 1980s the Solent was one

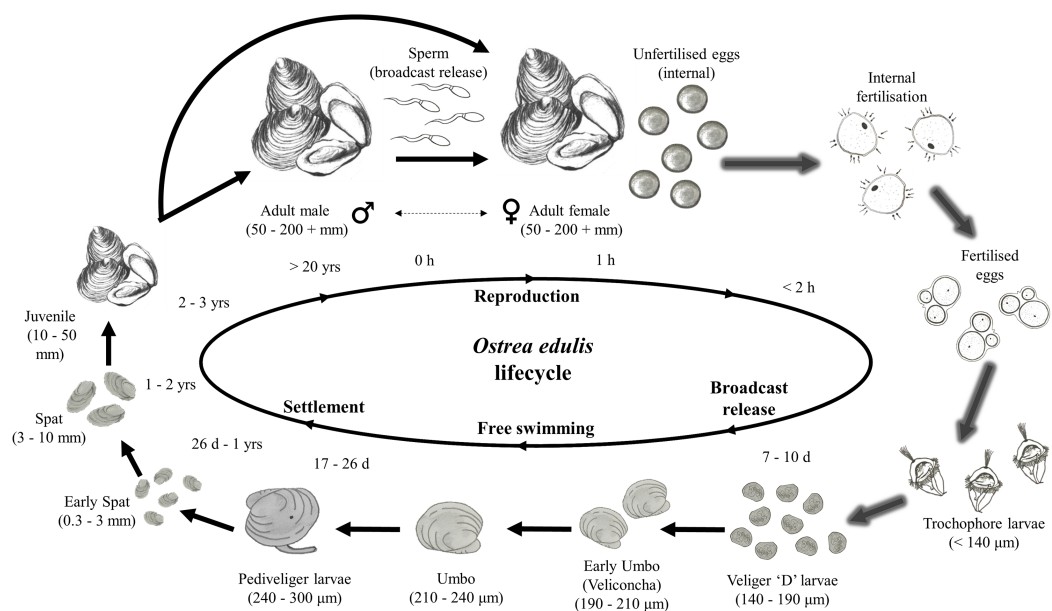

**Figure 1 Lifecycle of *Ostrea edulis*.** Arrows with glow effect indicate stages that occur internally within the female oyster pallial (mantle) cavity, plain arrows indicate stages that occur externally. Approximate sizes and timings are based upon information from *Hu et al. (1993)*, *Acarli & Lok (2009)*, *Food and Agriculture Organization of the United Nations (FAO) (2016)* and, *Loosanoff, Davis & Chanley (1966)*, *Pascual (1972)* and *Tanaka (1981)*, cited within *Hu et al. (1993)*. Images of life stages are not to scale.

of the larger remaining oyster fisheries in Europe, supporting 450 commercial vessels that landed 650–850 tonnes of *O. edulis* between Weymouth and Chichester during 1979–1980 and recorded seabed densities of 32/m$^2$ (*Key & Davidson, 1981*). The relative ease of access to their intertidal and coastal habitat facilitated the continued unsustainable extraction which, alongside a range of other environmental and anthropogenic pressures, led to chronic population decline (*Davidson, 1976*; *Key & Davidson, 1981*; *Tubbs, 1999*; *Vanstaen & Palmer, 2009*). At the turn of the 21st century annual stocks had decreased rapidly from 200 to 20 tonnes by 2011, which was mirrored in the decline of fishing licences issued, from 77 (2002/3) to 22 (2009/10) (*Kamphausen, 2012*). The Southern Inshore Fisheries and Conservation Authority (IFCA) closed the wider Solent completely to oyster fishing between 2013 and 2015 due to a failure of stocks to recover from a population crash in 2007 (Southern IFCA 2014 in *Gravestock, James & Goulden, 2014*). A lack of recovery that is in part due to the complex lifecycle of the species which is notoriously sporadic and comprises of multiple stages, including external sperm release, internal egg fertilization, vulnerable free-swimming larvae and larval settlement (Fig. 1). The unsustainable impact of overfishing is further compounded by multiple detrimental factors that are summarized in Fig. 2.

The disease Bonamiosis has severely impacted *O. edulis* populations. Caused by the intrahaemocytic protozoan parasite *Bonamia ostreae*, introduced into Europe via transplanted *O. edulis* seed from Californian hatcheries (*MacKenzie et al., 1997*), it has spread across Europe (*Grizel et al., 1988*; *Lynch et al., 2005*; *Culloty & Mulcahy, 2007*;
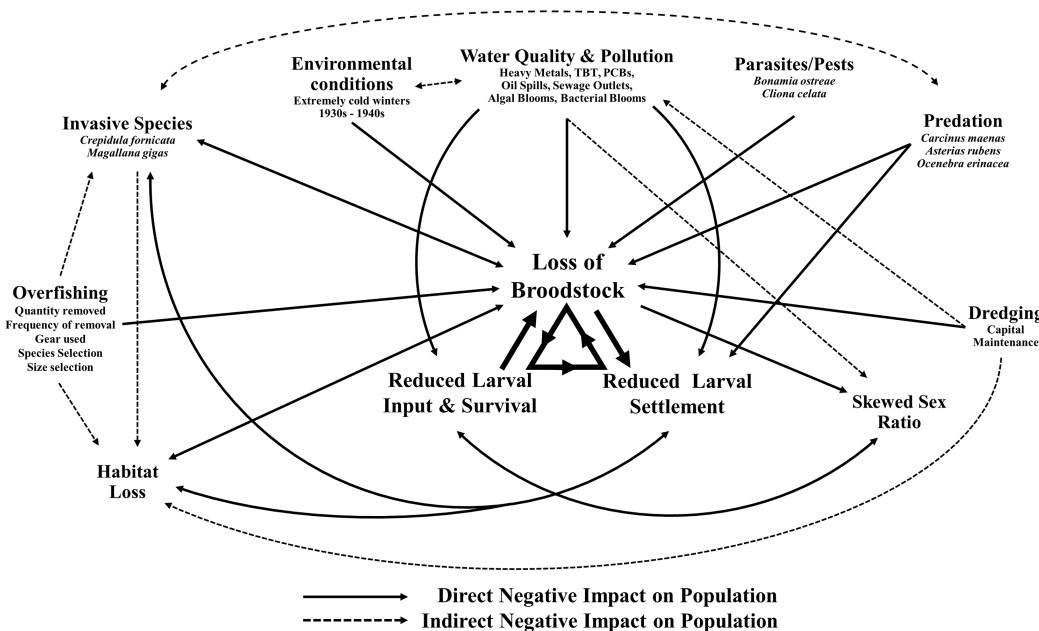

**Figure 2 Interactive effects adversely affecting *Ostrea edulis*.** The factors that are known to be adversely affecting *Ostrea edulis* populations within the Solent and their interconnecting relationships. Examples of the factors are shown where necessary.

*Lallias et al., 2008*) causing mass mortalities of up to 90% of localized populations (*Figueras, 1991*; *Cigarria, Fernandez & Lopez-Basanez, 1995*; *Laing, Walker & Areal, 2005*). The parasite becomes systemic within the host oyster, inducing physiological disorders that eventually become overwhelming, causing death (*Grizel, 1985*; *Grizel et al., 1988*). Although, it is thought that some resistance can arise through selective breeding strategies (*Baud, Gerard & Naciri-Greven, 1997*; *Culloty, Cronin & Mulcahy, 2001*; *Lallias et al., 2009*).

The presence of the invasive gastropod *Crepidula fornicata* is also a major concern across Europe (*Blanchard, 1997*), particularly in the Solent. The species was accidentally introduced with imports of *Crassostrea virginica* (*Dodd, 1893*; *McMillan, 1938*; *Hoagland, 1985*; *Utting & Spencer, 1992*; *Minchin, McGrath & Duggan, 1995*) and *Magallana gigas* (*Blanchard, 1997*) to Liverpool in the 1880s (*Moore, 1880* in *McMillan (1938)*) and the east coast and Thames estuary in the 1890s and early 1900s (*Crouch, 1893*; *Cole, 1915*). Despite claims that *C. fornicata* may increase macrozoobenthic communities in muddy sediments (*De Montaudouin & Sauriau, 1999*), its rapid expansion throughout many areas of the UK (*Orton, 1950*; *Chipperfield, 1951*; *Cole & Baird, 1953*; *Barnes, Goughlan & Holmes, 1973*; *Minchin, McGrath & Duggan, 1995*) and Europe (*Blanchard, 1997*, *2009*; *Davis & Thompson, 2000*; *Thieltges, Strasser & Reise, 2003*), including rapid colonization of oyster beds (*Crouch, 1893*), has serious ecological and economic impacts (see *Blanchard, 1997*). *Crepidula fornicata* has been shown to be detrimental to habitat suitability for juvenile fish (*Le Pape, Guerault & Desaunay, 2004*), suprabenthic biodiversity (*Vallet et al., 2001*), shell growth and survival of the bivalve *Mytilus edulis* (*Thieltges, 2005*). This invasive species is also attributed to habitat

modification, through the production of mucoidal pseudofaeces, whereby benthic substrata change from predominantly sandy to muddy with a high organic content that rapidly becomes anoxic and unsuitable for other species (*Streftaris & Zenetos, 2006*). This includes the native oyster through a reduction in suitable substrata available for larval settlement (*Blanchard, 1997*), hindering recruitment and potentially oyster restoration efforts on the seabed.

The shift toward a habitat dominated by *C. fornicata*, due to the decline of *O. edulis* and its accompanying biogenic habitat, is also of concern because of the loss of associated socio-economic benefits (*Grabowski et al., 2012*) and ecosystem services that other oysters species have been shown to provide (*Coen et al., 2007*). These services include, but are not limited to, increases in: biodiversity (*Wells, 1961*; *Zimmerman et al., 1989*; *Smyth & Roberts, 2010*), habitat complexity (*Bell, McCoy & Mushinsky, 1991*) nekton biomass (*Humphries & La Peyre, 2015*), fish abundance (*Coen, Luckenbach & Breitburg, 1999*; *Harding & Mann, 2001*; *Peterson, Grabowski & Powers, 2003*; *Tolley & Volety, 2005*) and nitrogen removal (*Piehler & Smyth, 2011*; *Kellogg et al., 2013*; *Smyth, Geraldi & Piehler, 2013*). It is recommended that further research is conducted to better understand these services in relation to *O. edulis*. The ecological significance and economic importance of this species within coastal temperate environments is also highlighted by its inclusion within the UK Biodiversity Action Plan (*UKBAP, 1999*; *Gardner & Elliott, 2001*), which describes the habitats or species of the UK and provides detailed plans for the conservation of native oysters. To add, further legislation encompassing the Native Oyster Species Action Plan (NOSAP) (*Hawkins, Hutchinson & Askew, 2005*) has been agreed. The NOSAP assesses the conservation status of *O. edulis* and its habitats, and outlines conservation priorities.

With the global decline of oyster reefs, beds and other habitats, oyster restoration efforts are growing in momentum and scope for widespread restoration in North East Atlantic marine protected areas has been proposed (*Fariñas-Franco et al., 2018*). Numerous restoration feasibility studies (*Laing, Walker & Areal, 2005*; *Shelmerdine & Leslie, 2009*; *Woolmer, Syvret & Fitzgerald, 2011*; *Gravestock, James & Goulden, 2014*; *Fariñas-Franco et al., 2018*) and restoration projects (*Roberts, Smyth & Browne, 2005*; *Eagling, 2012*: cited in *Gravestock, James & Goulden (2014)*; *Harding, Nelson & Glover, 2016*) have been, and are currently being, conducted within the UK. *Laing, Walker & Areal (2005*, 65–81*)* outlines many of the known previous attempts globally with *Zu Ermgassen et al. (2016)* outlining the management strategies for future conservation efforts.

Current baseline data is required to understand the fundamental ecological principles of distribution, density, growth, survival, reproduction and recruitment of the native oyster. The importance of such data is illustrated by *Christianen et al. (2018)* after recently discovering a 40-hectare mixed bed reef in an area where *O. edulis* was previously considered ecologically extinct. This study, provides data on the current distribution, abundance and demographic structure of *O. edulis* in the eastern Solent encompassing Portsmouth, Langstone and Chichester Harbours, sites intended for future oyster restoration efforts. The health status of each population is assessed using condition index (CI) and the screening of specimens for Bonamiosis infection. The change in fishery stock within the

local harbors is assessed as are the changes in abundance of *O. edulis* and *C. fornicata* within Chichester Harbour, by comparing current and historical datasets.

## MATERIALS AND METHODS

### Demographic assessment

Demographic population data from 2015 to 2017 were derived from oysters captured by commissioned dredge fishing at the beginning of the open fishing season of each stated year's fishery, with no selection for minimum landing size. All oysters were collected using ladder dredges in accordance with the local byelaw conditions for Portsmouth and Langstone Harbours (*Southern Inshore Fisheries and Conservation Authority, 2018*) and Chichester Harbour (*Sussex Inshore Fisheries and Conservation Authority, 2018*) IFCAs. Oysters were obtained from the entrance of Portsmouth Harbour (Hamilton Bank and Spit Bank, H+S, Fig. 3A) and within Chichester Harbour (Emsworth and Thorney Channels, E & T, Fig. 3A) during November 2015. Oysters from Langstone Harbour (Sinah Lake and Langstone Channel, S & L, Fig. 3A) were obtained during November 2016. Oysters from Langstone and Chichester Harbours were also obtained during 2017, but unfortunately, the fishers mixed these populations on landing. Live oysters were cleaned to remove epibionts and blotted dry before measuring. Measurements (Fig. 4A) for the maximum shell length, width and depth, as well as whole wet weight were recorded for a minimum of 700 oysters from each harbor. Maximum shell depth was not recorded for the first 500 Chichester and Portsmouth oysters, but was recorded for the final 200 individuals.

### Condition index and *Bonamia ostreae* prevalence

Oysters sampled for CI and *B. ostreae* screening were immediately frozen and stored at −20 °C. Condition index was performed to compare oyster populations from Chichester ($n = 24$) and Portsmouth ($n = 24$) according to the methodology in *Culloty, Cronin & Mulcahy (2004*, 45*)* with modifications, 105 °C for 24 h opposed to 60 °C for 48 h. The calculation used by *Walne & Mann (1975)* and *Lucas & Beninger (1985)* (cited in *Culloty, Cronin & Mulcahy (2004)* was used to determine CI:

$$\text{Condition index} = \frac{\text{Dry tissue weight}}{\text{Dry shell weight}} \times 100$$

In addition to the gill tissue samples taken before CI analysis, a further 24 oysters were selected at random from the landings of both harbors to determine the prevalence of *B. ostreae* infection. For each specimen a five mm section of gill tissue was removed with a sterile blade and genomic DNA extractions were performed using DNeasy® Blood & Tissue kits (product 69504; QIAGEN™, Hilden, Germany) following the manufacturer's tissue protocol (Qiagen, 2006). Quantification of the DNA extractions was conducted using a NanoDrop® 1000 Spectrophotometer (NanoDrop®, Thermo Fisher Scientific Inc., Wilmington, DE, USA). Species specific primers Oe fw_1/Oe rev_4 (*Gercken & Schmidt, 2014*) were used to amplify the cytochrome c oxidase subunit I (COI) gene from oyster DNA, as a positive control. Family and species specific primers (BO/BOAS; *Cochennec et al., 2000* and CF/CR; *Carnegie et al., 2000*) were used to amplify fragments of

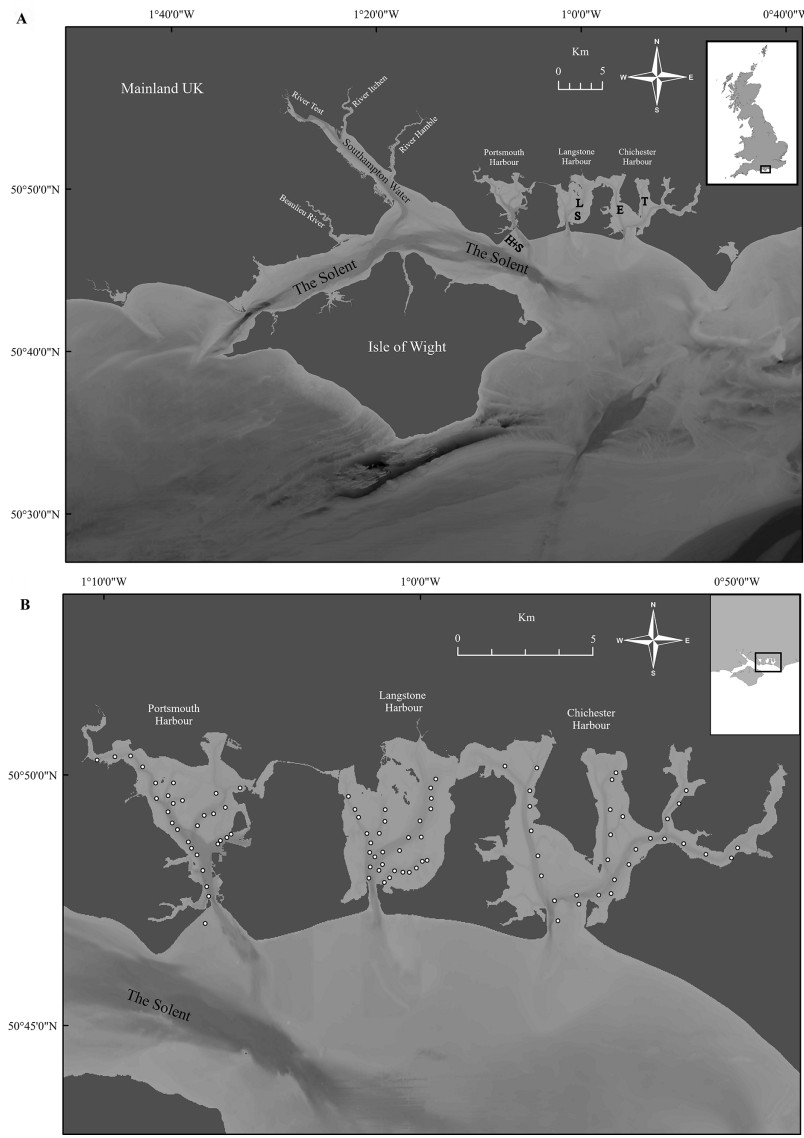

**Figure 3 Overview of Solent sampling locations.** (A) The wider Solent, showing the three harbors under investigation with locations of sample collection. (H+S) Hamilton Bank and Spit Bank, (S) Sinah Lake, (L) Langstone Channel, (E) Emsworth Channel, (T) Thorney Channel. (B) Benthic sample locations within Portsmouth, Langstone and Chichester harbours for the 2017 survey, three 0.1 m² samples were retrieved from each area marked by a ○ with areas selected to cover the maximum amount of each harbor within reason. Maps created using ArcMap software (http://desktop.arcgis.com/en/arcmap/).

the nearly complete small sub unit of 18S rDNA from all microcell members of the family Haplosporidiidae and specifically *B. ostreae*.

Gene amplification by polymerase chain reaction (PCR) was performed in 25 μl reactions using 1× DreamTaq™ PCR Master Mix (Thermo Fisher Scientific Inc., Wilmington, DE, USA), 0.2 μM each primer, 30–100 ng genomic DNA, and adjusted to the final volume with molecular grade $H_2O$. Reactions were conducted in a G-STORM 482—48 Well Multi Block Thermal Cycler (Gene Technologies Ltd., Essex, England)

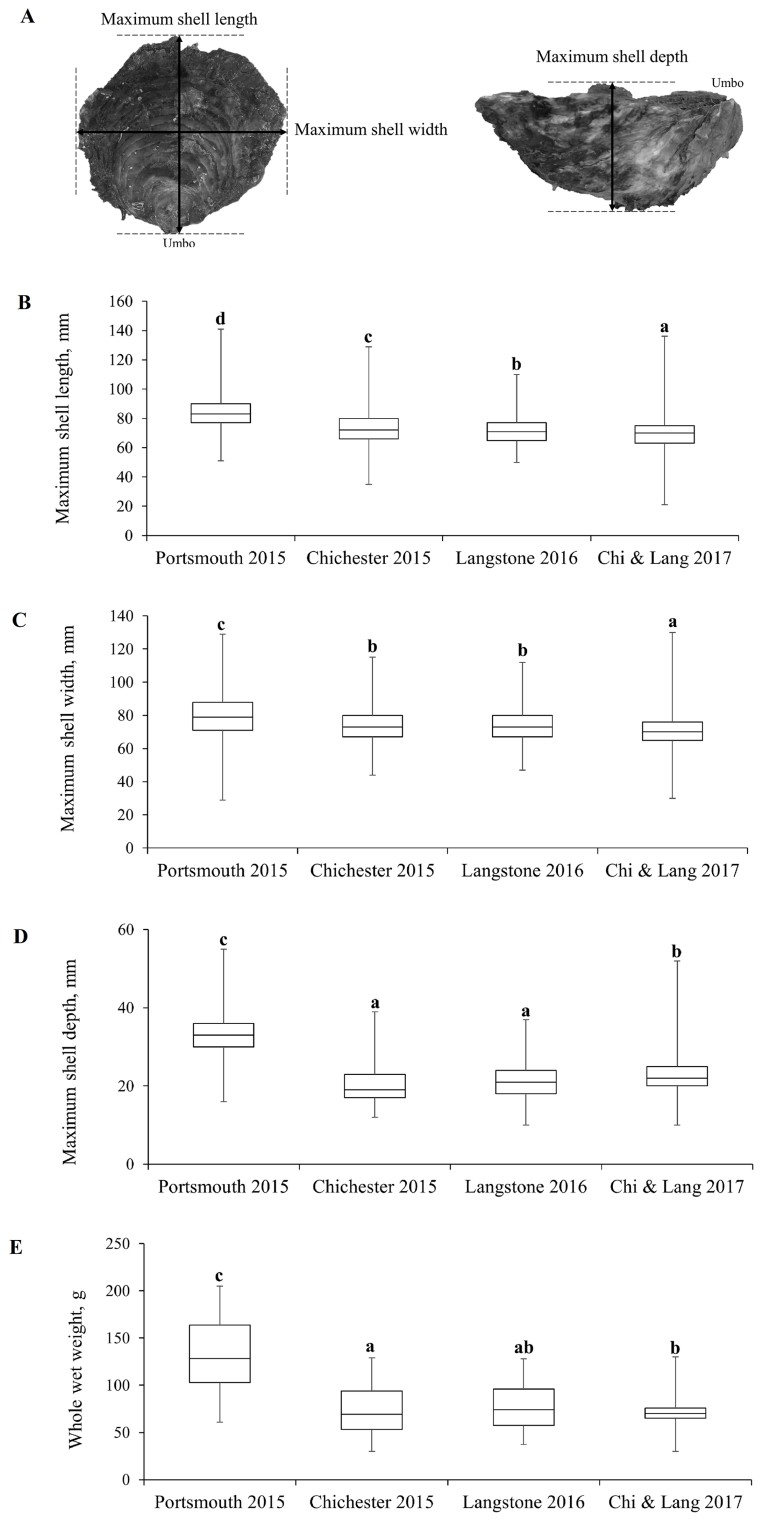

**Figure 4 Morphometric measurements of Solent oysters.** (A) Morphometric measurements recorded for *Ostrea edulis*. Box plots of morphometric parameters (Interquartile range, median and range of maximum shell (B) length, (C) width, (D) depth, mm and (E) whole wet weight, g) for *Ostrea edulis* populations ($n = 700$) across eastern Solent harbors during 2015–2017.

as follows. For Oe fw_1/Oe rev_4 primers: initial denaturation for 5 min at 94 °C, 35 cycles of amplification (1 min denaturation at 94 °C, 1 min annealing at 45 °C and 1 min extension at 72 °C) followed by a final extension at 72 °C for 10 min. For the BO/BOAS and CF/CR primers: initial denaturation for 5 min at 94 °C was followed by 35 cycles of amplification (1 min denaturation at 94 °C, 1 min annealing at 55 °C and 1 min extension at 72 °C) and then by a final extension step at 72 °C for 10 min.

Polymerase chain reaction products were visualized using 1% agarose gels (Fisher Scientific, Loughborough, UK) composed of 100 ml 1× Tris-acetate-EDTA buffer and 4 μl ethidium bromide (Sigma–Aldrich®, St. Louis, MO, USA). Using the GeneRuler™ 1 kb DNA ladder (Thermo Fisher Scientific Inc., Wilmington, DE, USA) for size quantification. Electrophoresis was run at 95 V for 1 h. Following this the samples were visualized by ultraviolet (UV) transillumination in a "VWR® Gel Documentation Smart Version system."

The oysters collected from Langstone Harbour ($n$ = 145) were analyzed, for presence or absence of *B. ostreae*, by the Centre for Environment Fisheries and Aquaculture Science (CEFAS) with all individuals screened using traditional histological methods (*OIE, 2003*). Any samples that showed evidence of infection were confirmed using single round PCR with BO/BOAS and *Bonamia* duplex primers. Any positive products were sequenced.

### *Ostrea edulis* and *Crepidula fornicata* benthic surveys: 1998 and 2017

The Chichester Harbour sample locations were identical for surveys completed in 1998 and 2017. The 1998 survey method (*Farrell, 1998*) varied from that in 2017, in that a box anchor dredge was used at 26 of the sample locations, three locations were hand dug at extreme low water spring tide and two locations were not sampled due to logistical reasons. Sample area was calculated as follows:

Dredge volume = 36 l = 36,000 cm$^3$
Mean thickness of sediment layer = 6 cm
Theoretical area sampled by full dredge = 36,000/6 = 6,000 cm$^2$ = 0.6 m$^2$.

The majority of the recent surveys occurred before the November 1st, 2017 in what would become the active fishery areas, other "closed" areas in Chichester Harbour were sampled after this date. A total of 30 or 31 locations were chosen within each harbor (Fig. 3B), with three replicate samples collected using a 0.1 m$^2$ Van-Veen grab at each location. For each sample all material collected was passed through a 6 mm square mesh box sieve, to remove excess sediment, and placed into individually sealed plastic bags when onboard the research vessel. Samples were then returned to the laboratory where they were rinsed and passed through a 6 mm square mesh box sieve for a second time to remove any remaining sediment to observe live organisms with ease. Total oyster (*O. edulis*) and limpet (*C. fornicata*) densities were recorded for each sample location.

Geographical position was assessed with a precision of 2 m using a Lowrance® Elite 7m GPS system. Distribution and abundance of oysters and slipper limpets were successfully surveyed at all sites within all harbors during the 2017 survey and 29 of the 31 proposed sites within Chichester Harbour in 1998.

**Table 1 Comparison of fishery population morphometrics.**

| Morphometrics | Portsmouth 2015 | Chichester 2015 | Langstone 2016 | Langstone & Chichester 2017 | Statistical difference between all group means across site/year |
|---|---|---|---|---|---|
| Length mm mean ± SE | 84.27 ± 0.44 | 73.85 ± 0.45 | 70.02 ± 0.36 | 69.96 ± 0.38 | $F_{3,2853} = 259$, $P \leq 0.001$ |
| Width mm mean ± SE | 79 ± 0.48 | 73.84 ± 0.39 | 71.02 ± 0.34 | 70.88 ± 0.36 | $F_{3,2853} = 89.8$, $P \leq 0.001$ |
| Depth mm mean ± SE | 33.1 ± 0.41 | 20.29 ± 0.33 | 23.03 ± 1.53 | 23.13 ± 0.24 | $F_{3,1853} = 305.9$, $P \leq 0.001$ |
| Weight g mean ± SE | 139 ± 2.34 | 79.42 ± 1.54 | 85.72 ± 1.52 | 87.22 ± 2.46 | $F_{3,2853} = 329.4$, $P \leq 0.001$ |

Note:
Fishery demographic data comparison between year groups across the three Eastern Solent harbors.

## Data analysis

All statistical analysis was performed in IBM® SPSS® Statistics 22 (IBM Analytics, New York, United States). Morphometric data (depth, width, length and weight) were analyzed for each separate parameter using one-way ANOVAs against year and site groups. Condition Index data were tested for homogeneity of variance using Levene's test and were found to be "normal" ($F_{1,46} = 0.9$, $P > 0.05$) and analyzed using a one-way ANOVA against location. Benthic survey data collected in 2017 for limpet densities were analyzed using an ANOVA general linear model (GLM) with harbor and site as independent variables. Oyster data were not suitable for statistical analysis within the ANOVA GLM. The mean densities of oysters and limpets within each harbor were compared against one another using paired student $T$-tests, as were the Chichester Harbour 1998 data. For comparisons between the 1998 and 2017 surveys, data were analyzed for each species using an ANOVA GLM with harbor and site as independent variables. FAO-ICALARM stock assessment tools II modal progression analysis of oyster length frequency distributions were used to identify distinct cohorts within each population. Minimum size class was specified at 15 mm with 5 mm size class intervals. Bhattacharya's method was used to determine initial decompose composite length-frequency distributions and refined using NORMSEP.

The univariate and non-parametric multivariate techniques using ordination from principle coordinate analysis (PCO) with data constrained in Bray Curtis similarity matrices were examined using PRIMER 6.1 (PrimerE Ltd: Plymouth Routines in Multivariate Ecological Research) to explore similarities with the relationship between the CI, maximum shell length and infection occurrence. PCO analyses were used for visualizing the results as an ordination, constrained to linear combinations of the variables. Similarities of the CI, maximum shell length and infection occurrences between localities were examined using PERMANOVA main tests and post hoc pairwise tests.

## RESULTS

### Population demographics

The interquartile range, median and range of the populations are shown in Figs. 4B–4E with statistically significant populations distinguished by lettering. There were statistically significant differences between group means across site/years (Table 1). There was a significant difference in the mean maximum length, width, depth and weight between

the oyster populations in Portsmouth and Chichester Harbours in 2015. There was no significant difference in mean width or depth between the 2015 Chichester and 2016 Langstone populations. Mean length and width has decreased across sites and years since 2015; the 2017 Chichester and Langstone population has significantly smaller oysters than all previous year/site groups.

The most frequent length size class recorded from the 2015 Portsmouth population was 81–90 mm (36%) in contrast to 61–70 mm (33.86%) in the 2015 Chichester population. The latter was also the most frequent size class in the 2016 Langstone (40.57%) and 2017 Chichester & Langstone samples (40.69%). The most frequent maximum shell width size class was 71–80 mm from the 2015 Portsmouth (30.43%), 2015 Chichester (30.43%) and 2016 Langstone (37%) populations and 61–70 mm in the 2017 Langstone & Chichester (43.85%) combined population. The demographic structure is narrower within the 2016 and 2017 populations sampled (Fig. 5). The NORMSEP modal progression analysis used to identify the number of cohorts, or age classes from the size frequency data confirmed the narrowing demographic structure and lack of recruitment cohorts (Table 2). The number and distribution of the cohorts suggest low levels of recent recruitment across all harbors and years. Three modes were estimated from the Portsmouth size class frequency data but dominated ($n = 652$) by the smallest cohort with an estimated mean of 84.57 ± 9.67 mm (modal mean ± SD) with a tail of low frequency larger size classes, effectively suggesting a single aged cohort. Three more evenly distributed modes were identified in the 2015 Chichester population with a smaller cohort of 71.73 mm ($n = 559$). The temporal trend demonstrates a decreasing population structure. Only two cohorts were identified in both the 2016 Langstone and 2017 Langstone & Chichester populations, the latter dominated almost entirely by a single cohort ($n = 743/757$) of 71.20 ± 8.78 mm. (modal mean ± SD).

## Condition index and prevalence of *Bonamia ostreae*

There was no significant difference between the CI of the Chichester, 3.3 ± 0.5 g dry wt (mean ± SE), and Portsmouth populations, 3.97 ± 0.5 g dry wt (mean ± SE), ($F_{1,46} = 0.9$, $P \geq 0.05$). The PCR provided 91 positive amplifications of the Oe fw_1 / Oe rev_4 positive control. Those that did not provide positive amplifications were discarded from the results. Those that provided positive amplifications showed that 46.8% of the Chichester oysters and 80% of the Portsmouth oysters, were not infected with microcell Haplosporidians or *Bonamia ostreae*. The remaining 53.2% of oysters from Chichester were infected, 32% tested positive for a microcell Haplosporidian other than *B. ostreae* and 21.2% positive for *B. ostreae*. In comparison to this, 20% of the Portsmouth oysters that were infected showed only positive amplifications for microcell Haplosporidians other than *B. ostreae*, with no evidence of *B. ostreae* found.

Incidences of bonamiosis within the Chichester population occurred across a range of different sized oysters. However, the majority (66.7%) occurred in oysters <82 mm in length, all with a dry tissue weight of <2 g. Incidence of infection with microcell Haplosporidians, other than *B. ostreae*, within the Chichester population occurred in

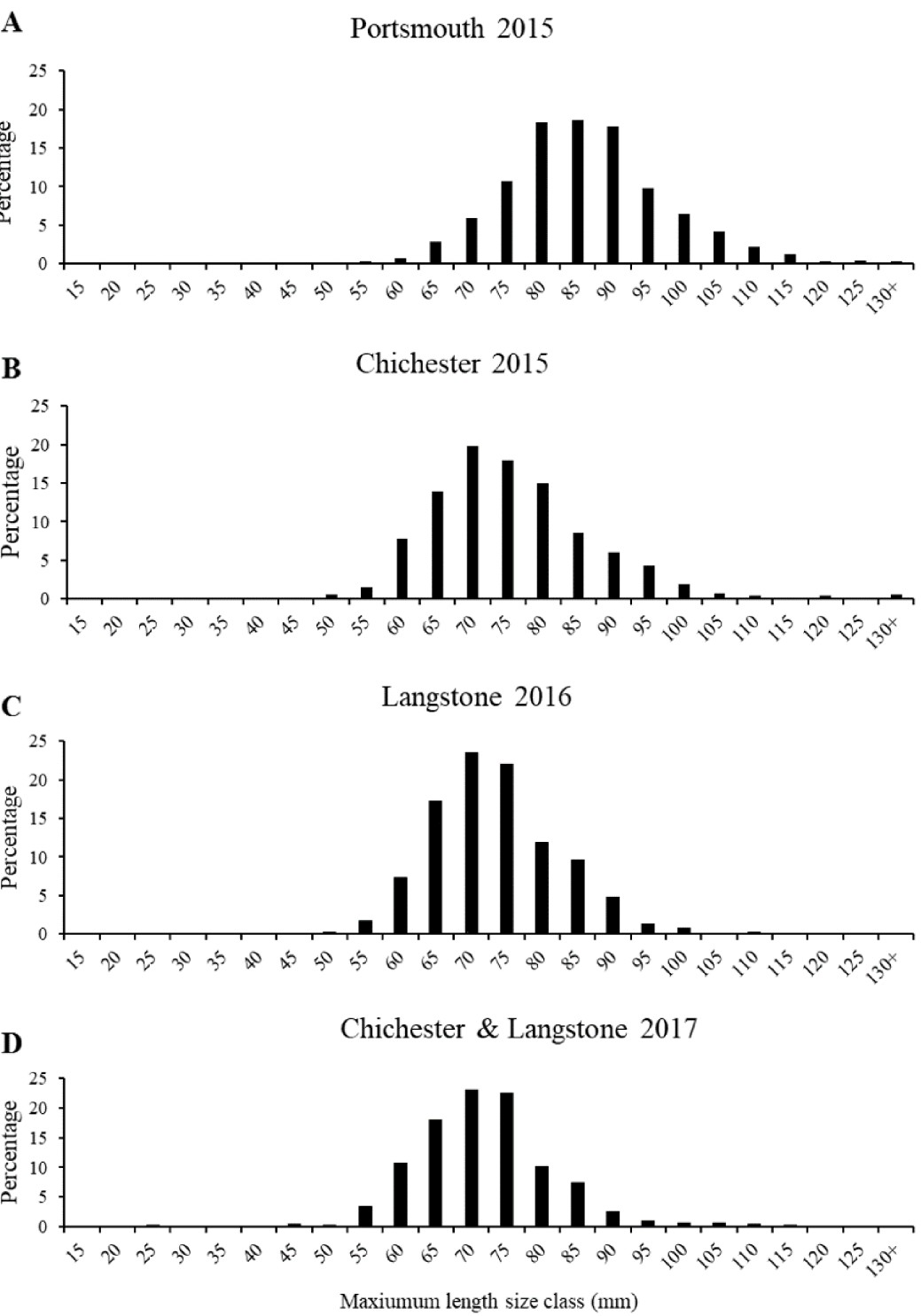

**Figure 5 Native oyster population demographics in the Solent.** Maximum mean length percentage frequency (with five mm intervals) of *Ostrea edulis* populations (*n* = 700 minimum) from (A) Portsmouth Harbour fishery population in 2015, (B) Chichester Harbour fishery population in 2015, (C) Langstone Harbour fishery population in 2016, (D) Chichester Harbour and Langstone Harbour mixed fishery populations in 2017. Refer to Fig. 3A for sampling locations of each fishery population.

**Table 2 Computed modal mean ± SD length (mm) cohort estimates.**

| Cohort/age class | Portsmouth 2015 | | Chichester 2015 | | Langstone 2016 | | Langstone & Chichester 2017 | |
|---|---|---|---|---|---|---|---|---|
| | Mean ± SD | $n$ | Mean ± SD | $n$ | Mean ± SD | $n$ | Mean ± SD | $n$ |
| 1 | 84.57 ± 9.67 | 652 | 71.73 ± 8.18 | 559 | 69.12 ± 6.18 | 400 | 71.20 ± 8.78 | 743 |
| 2 | 107.62 ± 6.26 | 43 | 89.35 ± 8.66 | 133 | 79.08 ± 8.76 | 300 | 103.33 ± 6.53 | 14 |
| 3 | 126.97 ± 2.65 | 5 | 124.19 ± 6.28 | 8 | | | | |

Note:
Computed modal mean ± SD length (mm) cohort estimates from length frequency analysis of all samples. $n$ = population.

oysters <87 mm in length, all with a dry tissue weight of <4 g. In comparison, the incidence of infection with microcell Haplosporidians, other than *B. ostreae*, within the Portsmouth population occurred in oysters between 70 and 100 mm in length, with dry tissue weights between 2 and 11 g. No relationship was observed between CI, maximum shell length and infection with *B. ostreae* ($F_{2,21} = 0.6$, $P \geq 0.05$).

A small proportion (4.1%) of the sample population from Langstone Harbour showed positive products and were sequenced, showing 100% homology to *B. ostreae* (KY296102.1) with those infected showing light to moderate levels.

## Densities of *Ostrea edulis* and *Crepidula fornicata* in 2017

During the survey, no oysters were found in Portsmouth Harbour, two were found in Langstone Harbour and one found in Chichester Harbour (Fig. 6A).

In contrast, *C. fornicata* was abundant in many areas, with mean harbor densities of 84.1 ± 24.5, 174.3 ± 34.5 and 306 ± 106 limpets/m$^2$ (mean ± SE) for Portsmouth, Langstone and Chichester, respectively (Fig. 6B). Both Langstone and Chichester Harbours contained significantly more individuals than Portsmouth Harbour. Even though Chichester contained more limpets/m$^2$ than Langstone, no significant difference was found ($F_{2,273} = 4.1$, $P \geq 0.05$). Significantly more *C. fornicata* were found across all three harbors compared with *Ostrea edulis* ($t$-value = 4.9, $P \leq 0.001$).

## Densities of *Ostrea edulis* and *Crepidula fornicata* in Chichester Harbour in 1998

In 1998 *O. edulis* was present in many areas of Chichester Harbour, with 14 out of 29 sites having oysters, the sites provided mean densities ranging from 0 to 88 oysters/m$^2$ and the overall harbor density was 8.0 ± 2.7 oysters/m$^2$ (mean ± SE) (Fig. 7A). *Crepidula fornicata* was present in 19 of the 29 sites within the harbor. Mean densities per sample site ranged from 0 to 1,224 limpets/m$^2$ and the overall harbor density was 181.2 ± 40.7 limpets/m$^2$ (mean ± SE) (Fig. 7B). There were significantly more *C. fornicata* than *O. edulis* ($t$-value = 4.9, $P \leq 0.001$).

## Long term data comparison of *Ostrea edulis* and *Crepidula fornicata* densities

In Chichester Harbour a significant decrease in *O. edulis* density was observed between 1998 and 2017, from 8.0 ± 2.7 to 0.1 ± 0.1 oysters/m$^2$ (mean ± SE) ($F_{1,172} = 19.3$, $P \leq 0.001$)

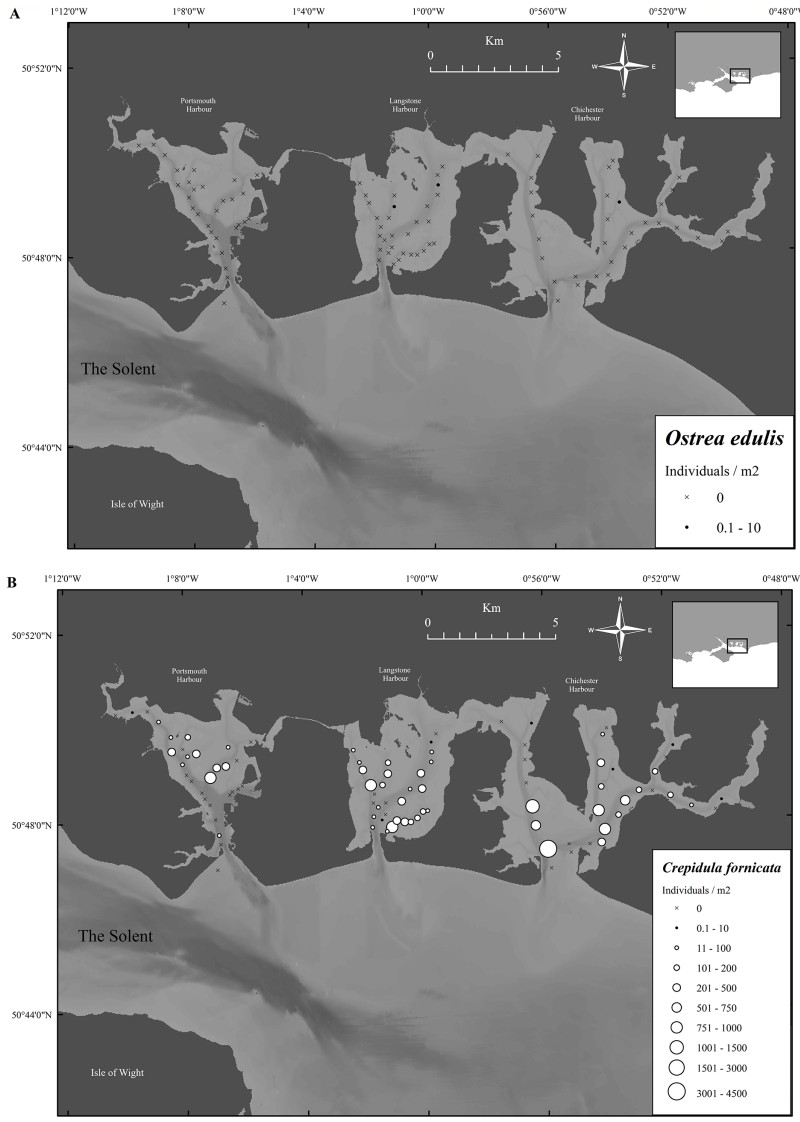

**Figure 6 Native oyster and slipper limpet distributions within the Solent harbors during 2017.**
(A) Mean densities of *Ostrea edulis* at the sampling locations in Portsmouth, Langstone and Chichester harbours, 2017. (B) Mean densities of *Crepidula fornicata* at the sampling locations in Portsmouth, Langstone and Chichester harbours, 2017. Maps created using ArcMap software (http://desktop.arcgis.com/en/arcmap/).

(Fig. 7C). In comparison, a significant increase was observed for *C. fornicata* between 1998 and 2017, from $181.2 \pm 40.7$ to $306 \pm 106$ limpets/m$^2$ (mean $\pm$ SE) ($F_{1,142} = 10.4$, $P \leq 0.01$) (Fig. 7D).

## DISCUSSION

In addition to the comprehensive stock assessment conducted by the Southern IFCA (*Southern Inshore Fisheries and Conservation Authority, 2017*) the data presented here are essential for determining the relative distribution and benthic composition of oysters and slipper limpets to provide a baseline status of the Eastern Solent. The information

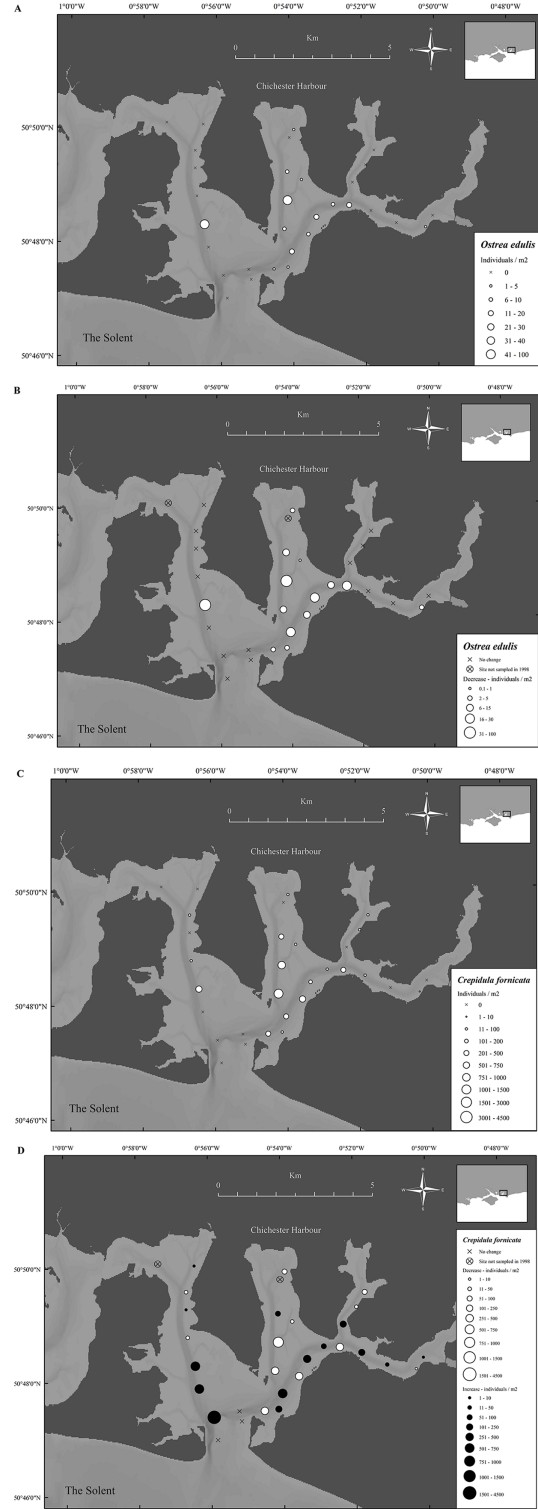

**Figure 7 Temporal change in native oyster and slipper limpet distributions over 19 years.** (A) Densities of *Ostrea edulis* in Chichester harbour, 1998. (B) Change in *Ostrea edulis* densities in Chichester Harbour from 1998 to 2017. (C) Densities of *Crepidula fornicata* in Chichester harbour, 1998. (D) Change in *Crepidula fornicata* densities in Chichester Harbour from 1998 to 2017. Maps created using ArcMap software (http://desktop.arcgis.com/en/arcmap/).

provided can be used to inform restoration initiatives and also used to determine the future success of restoration activities proposed the Solent (*Harding, Nelson & Glover, 2016*). The narrowing of fishery landing sizes combined with the significant decrease in the abundance of the ecosystem engineer, *Ostrea edulis*, within Chichester, Langstone and Portsmouth Harbours indicated that these populations of native oysters are not recovering, recruiting or present in reproductively relevant densities.

The long-term decline of *O. edulis* standing stock is caused by a combination of chronic overfishing, poor water quality (*Environment Agency, 2016*) and disease, which suggests a loss of the ecosystem services that biogenic oyster habitats are able to provide. As demonstrated by studies of various oyster species, such a loss will likely have a profound negative impact on biodiversity, benthic community structure, trophic pathways and water quality across the Solent as it is highly probable that oysters and oyster reefs are universal in terms of ecosystem services provision (*Lenihan, 1999*; *Jackson et al., 2001*; *Peterson, Grabowski & Powers, 2003*; *Tolley & Volety, 2005*; *Smyth & Roberts, 2010*). The continued expansion of the invasive, non-native and highly successful gastropod, *Crepidula fornicata*, has been facilitated by the decline in oysters, impoverished habitat, global shipping movements and a prolific lifecycle (*Richard et al., 2006*). The presence of such high densities highlights the impoverished state of the habitat, now a silty mud-dominated benthos which presents a barrier to the restoration of oyster-related benefits that previously once existed (*Korringa, 1946*; *Barnes, Goughlan & Holmes, 1973*; *Erhold et al., 1998*; *Thouzeau et al., 2003*; *Streftaris & Zenetos, 2006*).

First sighted in oyster ponds in Bosham, "Portsmouth Bay," during 1913 (*Cole, 1952*), and later in the wider Solent in 1930, *C. fornicata* spread west during the 1940s (*Blanchard, 1997*). By the 1970s the Solent was almost characterized by *C. fornicata* dominated associated macrofauna. However, *O. edulis* still occurred with a 19% frequency, but greater in the West (45%) than the East Solent (9%), (*Barnes, Goughlan & Holmes, 1973*). This persistent and increasing dominance of *C. fornicata* since the 1970s is of serious concern for the natural recovery of *O. edulis,* particularly the extremely high densities of over 4,000 m$^2$ within Chichester Harbour which demonstrates the ecological carrying capacity of inshore waters for this invasive species. The extent to which *C. fornicata*, at high densities, competitively excludes *O. edulis* should be tested.

The presence of *C. fornicata* not only negatively impacts the broodstock oyster population but *O. edulis* larvae could also be subject to substantial predation (*Korringa, 1951* in *Pechenik, Blanchard & Rotjan, 2004*) and competition. Oyster larvae attempting to settle will suffer competition for food and space from high *C. fornicata* larval densities (*Fitzgerald, 2007*) which form a cohesive calcareous shell-mud where they become dominant in contrast to hard substrate preferred by oyster larvae (*Smyth et al., 2018*). The reduced availability of suitable settlement substrate for *O. edulis* larvae due to both the levels of mucus pseudofaeces (*Blanchard, 1997*) generated by such high densities of *C. fornicata* and the lack of conspecific shell substrate is a major concern. A reduction in preferential substrata will be further compounded by competition for food arising during the overlapping breeding periods of both species, with *C. fornicata*

spawning two to four times between February and September (*Richard et al., 2006*) and *O. edulis* typically spawning once between May and August (*Hayward, Nelson-Smith & Shields, 1996*). The reproductive cycle of *O. edulis* is also relatively complex in relation to other oyster species and that of *C. fornicata* which is sexually mature within 2 months (*Richard et al., 2006*). In comparison, *O. edulis* is not usually mature until individuals reach 3 years old (*Roberts, Smith & Tyler-Walters, 2010*).

Again this is problematic as the remaining natural populations of mature oysters, within the sampled harbors of the Solent, have decreased significantly in size and abundance over a short time frame. This smaller, less mature and narrowed demographic population will negatively impact spawning potential. For example, a decrease in mean size from 80 to 70 mm, similar to the decline observed from 2015 to 2017, would result in a reduced output of 260,000 larvae per reproductive female (*Walne, 1974*). When applied to a fishery, characterized by a skewed sex ratio (6:1 male:female sex ratio (*Kamphausen, Jensen & Hawkins, 2011*)), reproductive and recruitment success will be severely impacted. This is of great concern and as a conservative estimate, 85% of the 2017 Langstone & Chichester population will be above the minimum landing size for the 2018/19 fishing season and therefore at high risk of being extracted to the point of functional extinction (*Beck et al., 2011*). To put this in context, in 1973 only 22% of *O. edulis* population in Stanswood Bay were of marketable size >70 mm (*Barnes, Goughlan & Holmes, 1973*), and it was this population that was thought to feed the boom of the Solent fishery in the late 1970s and 1980s.

This risk of extirpation at current fishing levels seems particularly high as there is little or no sign of recruitment cohorts for the sampled harbors. The last successful recruitment, estimated from size (*Richardson et al., 1993*), was approximately 5–6 years ago in 2012 for the Langstone & Chichester populations. The smallest 2015 Portsmouth cohort with a mean maximum length of 84 mm suggests this is the aged remnant population from a successful spatfall 8–10 years ago, approximately in 2008. The morphometric data reveals a disjunct population structure over microgeographic scales within the Solent, particularly between the Portsmouth and Chichester Harbours in 2015. This could be attributed to the re-stocking that took place as part of a small-scale restoration project in Chichester during November 2010 (*Vause, 2010*; *Eagling, 2012* cited in *Gravestock, James & Goulden, 2014*; *Marine Environmental Data & Information Network, 2016*), which aligns with the estimate from the demographic cohorts. However, it is concerning that the demographic data showed a lack of recruitment to the seabed in all three harbors despite the previous two years of fishery closure and a reduced fishing season since 2015.

When planning and managing projects to restore such populations, disease control is of utmost importance. The prevalence of *Bonamia ostreae* within the Chichester population in this study has increased from previous years (1993–2007 average 12.1% (*Laing et al., 2014*)) and is in agreement with the findings of *Eagling (2012*, cited in *Gravestock, James & Goulden, 2014)*, who reported disease prevalence of 25–35%. This increase could be attributable to the mortality of many of the re-laid oysters (at a density of 20/m$^2$) within the harbor observed by Jensen (personal communication,

A. Jensen, 2014 in *Gravestock, James & Goulden, 2014*) and indicate that this area may be susceptible to future outbreaks of Bonamiosis.

In contrast to this, the parasite was completely absent within the Portsmouth population decreasing from an average of 5.6% (1993–2007 (*Laing et al., 2014*)). This result is encouraging with the apparent absence of *B. ostreae* from a population previously exposed and subject to the parasite and could be indicative of resistance. Again, in contrast to the increase within Chichester, a reduction in prevalence was recorded within the Langstone population (9.1% mean 1993–2007 (*Laing et al., 2014*)). This suggests that although the three harbors are all interconnected the hydrodynamics of the area appear to prevent dispersal of the parasite in a westerly direction either in the water column or via larval transmission (*Flannery, Lynch & Culloty, 2016*). This is supported by the distinct demographic structure of the Portsmouth population, however, given the exceptionally low population densities this could be a density dependent phenomenon.

It is clear that the *C. fornicata* dominated benthos of the Solent harbors is in a highly altered state and without significant intervention or disturbance, presents a barrier to the return of the native oyster *O. edulis* along with the biogenic habitat and associated biodiversity it provides.

## CONCLUSION

The low standing stock of *Ostrea edulis*, coupled with a benthos dominated by high densities of *Crepidula fornicata*, the presence of *Bonamia ostreae* and continued fishing pressure are significant barriers to self-sustaining native oyster populations within the Solent. Based on the status of *O. edulis* in the commercially fished areas of the Solent presented here, active management of the seabed is recommended to (1) control the extent and spread of *Crepidula fornicata*, (2) provide suitable settlement substrate for *O. edulis* larval recruitment and (3) establish a protected *O. edulis* broodstock population in all commercially fished Solent harbors, in agreement with *Fariñas-Franco et al. (2018)*.

This paper highlights the importance of understanding local population structures and disease prevalence over relatively small geographic scales and reinforces the need for relevant and comprehensive baseline data to underpin *O. edulis* restoration practices. All the results highlight the impoverished state of the native oyster, in what was once a substantial population, thus the need to restore the species to the area.

### Funding

This study was primarily funded by the research budget assigned to Dr. Joanne Preston by the University of Portsmouth as well as from a University of Portsmouth and Blue Marine Foundation match funded PhD scholarship. Additional funding was also received from the Solent Forum. The funders had no role in study design, data collection and analysis, decision to publish, or preparation of the manuscript.

## Grant Disclosures

The following grant information was disclosed by the authors:
University of Portsmouth as well as from a University of Portsmouth.
Blue Marine Foundation match funded PhD scholarship.

## Competing Interests

Morven Robertson is employed by the Blue Marine Foundation. Simon Harding was employed by the Blue Marine Foundation at the time of data collection and manuscript drafting.

## Author Contributions

- Luke Helmer conceived and designed the experiments, performed the experiments, analyzed the data, contributed reagents/materials/analysis tools, prepared figures and/or tables, authored or reviewed drafts of the paper, approved the final draft.
- Paul Farrell conceived and designed the experiments, performed the experiments, contributed reagents/materials/analysis tools, approved the final draft.
- Ian Hendy analyzed the data, contributed reagents/materials/analysis tools, authored or reviewed drafts of the paper, approved the final draft.
- Simon Harding authored or reviewed drafts of the paper, approved the final draft.
- Morven Robertson performed the experiments, contributed reagents/materials/analysis tools, approved the final draft.
- Joanne Preston conceived and designed the experiments, performed the experiments, analyzed the data, contributed reagents/materials/analysis tools, prepared figures and/or tables, authored or reviewed drafts of the paper, approved the final draft.

## Data Availability

The raw data are available in the Supplemental Files.

## Supplemental Information

Supplemental information for this article can be found online at http://dx.doi.org/10.7717/peerj.6431#supplemental-information.

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
