# Peer review of "Active management is required to turn the tide for depleted Ostrea edulis stocks from the effects of overfishing, disease and invasive species"

_PeerJ, doi:10.7717/peerj.6431_

## Round 0.1 · original submission · Major Revisions

The reviewers have provided a number of concerns, comments, and suggestions that require major revision. At the moment the manuscript is not ready for publication. I encourage the authors to address all the comments provided.

Reviewer 1 ·

Basic reporting

There are a few very minor structural and editorial errors - I have included them in my general comments

This is an extensive piece of study - with many researched references. There is some concern that many are used inappropriately - where reference is made to the study of intertidal reef forming oysters when we do not yet know if these studies are relevant to sub tidal European flat oysters.

Experimental design

I think the experimental design is sound and the ideas that are being tackled profoundly important, relevant and timely. I have a major concern on the use of a small grab sample with low replication to determine the current oyster stock densities. I have discussed this extensively in my general feedback - but I do not see any apriori design in the approach to formally test whether there is a competitive interaction between crepidula and the oysters. We need more information on the methods of the 1998 study to know whether the historical and current oyster densities are comparable. If these have similar methods I think it can be concluded that there has been a decline in oysters - but acknowledging that the method is inappropriate to provide an accurate density estimate.

There is an excellent approach taken to the demographics and disease dynamics with very clear policy implications - this is good work that is supported by the data.

Validity of the findings

See above and general feedback.
I have a significant concern about the density estimation methods and the jump to stated conclusion that competitive exclusion is occurring.The conclusion as stated is not well supported by the results.

For sure competitive exclusion could be occurring but no data has been provided or formal test undertaken to show this. This is easy to remedy by using different language in communicating these ideas - communicating them as hypotheses and not as evidenced outcomes. I have some concerns about the data analysis - these are easy to fix and some suggestions have been provided. My biggest problem is lack of reporting of why transformations were chosen and whether they worked - lets see the data distributions - based on the descriptions in the text it appears GLMs are more appropriate and multiple testing needs to be dealt with or justified.

Additional comments

Line 89 What is the expected recovery since 2007 – would you have expected recovery in this time period?
Line 94 and 96 italicise species name
Line 109 - insert probable or likely before unsustainable
Line 127 – so the fishery remained open after the crash until 2013?
Line 131 – is this expert opinion or do you know this? Do you know the production to make this statement? Otherwise state - likely exceeded
Line 134 – Is this referring to all species or only in the context of flat oysters. In the context of flat oysters – can you differentiate between reefs and beds – I see later you do exactly this.
Line 157 – did Blanchard confirm or measure consequences – or discuss what could be consequences? In 2009 Blanchard discusses this again – but also finds that despite significant colonisation by Crepidula – they remain co-associated with extensive flat oyster beds. But there are clear level of effects based on %cover and density. I will ask about this later – can you relate your data to the % cover in the French study?
Line 163 – again – potentially inducing difficulties with recruitment.
Major point – From Line 167 – It is important to not (yet) overegg the potential ecosystem services that could be gained from O edulis restoration. The Cohen paper is brilliant – but it is not relevant to O.edulis. It should be stated that there are potential O.edulis ecosystem services and that here too knowledge gaps remain. For the following statements made through to 173 – please differentiate which oyster species each of these studies refer to. Many appear to be about intertidal reef forming oysters.
Line 178 –of different species – or only of oysters as per the acronym?
Line 180 – decline of reefs, beds and other habitats?
Line 190 – to link with the previous paragraph – this study is being done for baseline data to then embark on restoration? Are you really measuring ecological resilience?
Line 191 – the increase is the result, no? The “change” in abundance….
Line 194 – can you directly link any change in abundance to causal change?

Line 224 – this sentence should come later after you say what this paragraph is about? This is for the B.ostreae determination so say that at the beginning of the paragraph.
Line 260-279 – with important consequences for the discussion on using grabs below – you don’t state what the methods were in 1998 – did it use the same sized grad, differ grabs, more grabs or dredge to estimate density? Are the two methods comparable? The link to the Farrell study is not working. This is crucial – and would help justify your use of these methods again – if the 1998 study determined its density estimates with the same method – small grabs with 3 replicates
Major concern- Line 271 – we need to discuss the use of a grab to determine the densities of oysters. The vast majority of your sites had a measured density of zero – other than three sites. You already present information that there are oysters in these harbours. Your grab samples 0.1m/2 which would – on average – need to take 10 replicates of low density area to accurately estimate densities of 1-2 oysters per m2. But you state your replication at each sample area is 10. Blanchard used three grabs but of 0.25m2. These concerns increase if we assume most areas won’t have homogenous densities but very patchy distributions of oysters. For example in a 10x10m patch with a density of oysters of 1m2, if a 2x2m2 sub-patch contains 25 oysters the remaining 1m2 subpatches will have >1 oyster m2 – making the area very difficult to sample by grab. In the abstract you state clearly that you cannot differentiate between any of the densities and zero – and I do not believe this reflects the density in the harbours.
That on line 373 you state you only found 3 oysters in your entire survey suggest to me you have to consider dropping a calculation of density from this. This does not apply to the crepidula – where given the abundance you are encountering the grab is appropriate.
Are there no associated data from the 2015 dredges to estimate density?
Line 283 – Is Manova not more appropriate here – if not then correct for multiple testing.
Line 287 (and 292) – Why a square root transformation? Did it normalise the data? You need to confirm that the transformation worked - or whether there is otherwise good reasons to proceed with this model. Are GLMs nor more appropriate in this context – you need to better justify model choices. Potentially include species as a dependent variable – then you reduce the number of models you are building. What is different between the benthic survey data and density data? Given the number of oyster you found – the density data is likely very zero inflated and anova is in no way appropriate (but see points of method suitability – it is unlikely your method has the statistical power to detect any real differences).
Line 289 – what do you mean by densities were analysed against total abundance? Again are the 1998 and 2017 surveys comparable?
Line 388 – The use of the term haul suggests the 1998 surveys were done by dredge? 88/m2 – that is huge – please confirm if correct and again how this was measured – does this include spat?

Line 405 –You have undertaken a point sampling study with a grab of a size only suitable to measure high density organisms. That being said – you could determine what the minimum effective density your grab can ascertain with 3 replicates and then find the maximum density that could be in your study area?
Line 409 – you have provided more evidence that credpidula is an ecosystem engineer than O.edulis. I want to back you on this this but I think it is important that you have evidence that O.edulis is an engineer either in this study or from others.
Line 412 – this is important and I think you can state this – as densities are clearly low. Whether there are sub patches you have not sampled that are at higher density could affect this. Clearly if there are 55-90mm oyster throughout the harbours in 2015-2017 – that must mean they have recruited to the population both just before and after 2007. Is the lack of smaller oysters in these samples due to the dredge mesh size – 45mm?
Line 414 – there are some big claims here but do you know or have you evidenced that this is the reason for the crepidula expansion. Dredging will spread it around for sure – dosnt Blanchard discuss the spread of crepidula.
Major point – Line 416 - you have no formal test of competition between the shellfish. You don’t have enough variation in your data (which is due to the grab not that it may not exist) to test whether the decline in O.edulis is highest where Credpidula gains are greatest. There is no reasonable way to conclude the O.edulis is suffering from competitive exclusion in the Solent. This is a viable hypothesis – but I cannot support this statement in its current form.
Line 419 – is it appropriatre to link the Jackson study to a “likely” decline in water quality. Was there a decline in water quality in 2007-2008 after the crash? So many other multistressors in the UK coastal environment are likely to affect this – the biomass of crepidula – also able to filter feed – could be taken into account. I am not comfortable with repeating ideas and hypotheses as likely effects – taking from studies of extensive intertidal oyster reefs and using this to predict effects of O.edulis loss.
Line 421 – Korringa 1946 is not in the reference list. Does this reference find a 3D oyster bed/reef like habitat in the Solent? For sure we know/suspect that many UK estuaries have increased mud and subtidal mud coverage – this includes where crepidula are and are not present in high densities. So habitat change is not all about crepidula – but it is certainly a part of it.
Line 423 – again you are jumping to major conclusions here about the implications of credpidula when you have not measured ecosystem function. You have stated that the results are mixed with more studies suggesting negative effects than potential positive effects – although those positive effects are more likely in a mud dominated low species diversity estuary.
Can you formally test any change in oyster density with change in crepidula density - a regression maybe? I missed this if you did it already.

Line 439 – 440 – this is good on competition – it is here you should suggest that where crepidula at highest density there is a possibility of competitive exclusion – a hypothesis that could be tested.

456-501 – this is brilliant! – a very good policy and impact relevant discussion that is supported by the data presented.

Line 509-511 – we have no idea if O.edulis – at even reasonable densities of 5m2 - has the ability to do this. Again I stress the danger of taking work on intertidal very high density reef forming oysters and placing that in the context of lower density O.edulis.
Figure 4 does not have an explanatory legend or state what the summary statistic being displayed is. It appears to be median, IQR and range but you are analysing this data using an ANOVA that tests for differences between means? Present mean and SD if you want to provide descriptive information or Mean and 2 standard errors if you want to convey significant differences between sites.
Figure 5 – y axis spelling error

·

Basic reporting

The manuscript is reasonably well written and the standard of English is good. However, it is over-length, cluttered in places and the structure could be improved. The authors should revisit some of the material throughout and stream-line to match more clearly defined USPs / hypotheses (see comment below).

The article conforms to professional standards of courtesy and expression.

Literature references are sufficient to provide an introduction to the field and appropriate background/context. The authors should use an additional recent publication that might help in lines 181 and 184:

• Fariñas-Franco JM, Pearce B, Mair JM, Harries DB, MacPherson RC, Porter JS, Reimer PJ, Sanderson WG. Missing native oyster (Ostrea edulis L.) beds in a European Marine Protected Area: Should there be widespread restorative management?. Biological Conservation. 2018 May 31;221:293-311.

The structure of figs and tables is of a professional standard. Figure 1, has a passing reference to it in the Introduction and the case for it’s value to the m/s is therefore not well made. Given the title of the manuscript, a graph that illustrates the decline in the Solent oyster fishery may well be more appropriate.

Figure 3 and 5 have dark text on a dark background such that a better contrasting combination should be considered. There are also inconsistencies in labelling such that Southampton Water is in CAPS whilst “The Solent” and ”…Harbours” are not. Font sizes also differ between “A” and “B” panels in Figure 5.

Raw data is shared.

Lines 90 to 96 do not appear to be well placed at this point in the Introduction: They go some way to articulating the importance of the work but might be better merged with the last paragraph in the Introduction. The last paragraph of the Introduction should be clarified. Baseline data are useful but the justification for the study in lines 188-193 needs to much more clearly articulate hypotheses or study aims. Clarifying this paragraph will far better define the relevance of the various Results and Figures that follow.

Experimental design

The m/s presents original primary research within the Aims and Scope of the journal. Research questions need to be better defined (above) but broad content appears to be contemporary, relevant and meaningful. How the research fills an identified knowledge gap needs further clarification in that the research questions need more exact articulation.


The title of the m/s suggests that Ostrea edulis has been outcompeted by Crepidula fornicata. The study shows that when two studies in 1998 and 2017 are compared, Ostrea edulis has been replaced in its former locations by Crepidula fornicata. It is possible that the authors are correct but competition is not demonstrated and therefore “competitive exclusion” is a ‘stretch’ for the title (disease, habitat destruction and life history are also cited by the authors as contributing factors to decline of Ostrea edulis). Similarly, Ostrea edulis is shown to have declined but the case for overfishing needs to be more clearly articulated since there is no evidence presented that establishes a causal link (see above comments).

Rigorous investigations were performed to high technical & ethical standards. However, the relevance of morphological data is not entirely clear (see comments on hypotheses and study aims) and the authors should therefore remove these data and the ensuing Figure 4, include as supplemental information, or better justify their inclusion. If the morphometric data are retained the authors should consider the implications identified for Modiolus modiolus in:
• Brash JM, Cook RL, Mackenzie CL, Sanderson WG. The demographics and morphometries of biogenic reefs: important considerations in conservation management. Journal of the Marine Biological Association of the United Kingdom. 2017 May:1-0.
• Fariñas‐Franco JM, Sanderson WG, Roberts D. Phenotypic differences may limit the potential for habitat restoration involving species translocation: a case study of shape ecophenotypes in different populations of Modiolus modiolus (Mollusca: Bivalvia). Aquatic Conservation: Marine and Freshwater Ecosystems. 2016 Feb;26(1):76-94.



Methods are described with sufficient detail & information to replicate.

Validity of the findings

With some re-working, this study is of interest to an emerging international audience in marine environmental restoration. Data is robust and statistically sound.
Conclusions need to be better linked to original research questions (above).
Speculation should be identified as such.

Additional comments

Check italics of species names throughout
Portsmouth, Langstone and Chichester Harbours are inconsistently Capitalised throughout. As place names they should, presumably, be u/c.
Line 224: Insert ‘were’ before “randomly”
Line 225-226: sentence in note form
Line 233-234: This sentence indicates the purpose of the paragraph. Open paragraph with indication of this purpose.
Lines 249-252: re-visit
Line 254: Start sentence with words
Line 255: What were they analysed for?
Line 260: Here and elsewhere you refer to “1998-2017” rather than ‘1998 and 2017’. This gives the impression there is a time series rather than two surveys in the respective years. Care how this is presented throughout.
Lines 270-279: Was the same methods used in 1998 and 2017? 1998 is not mentioned.
Lines 275 and 286: “Ostrea edulis and Crepidula fornicata” is referred to then “oyster and limpet”. It would help the reader to know that Ostrea edulis is an oyster and Crepidula fornicata is a limpet at the first mention in this section.
Lines 314-330: Better in table-form
Lines 334-335: Common recruitment event/source (?) [both may be connected to elsewhere and not each other…. Unless there is additional evidence to support the assertion]. Either way, the Discussion section is probably a better place for this speculation.
Line 340-342: Improve clarity here.
Line 345: consistency with CAPs
Lines 395-396: The collapse of the fishery could be usefully evidenced in the present work as a foundation for the ‘before-after’ comparison implicit in the 1998 and 2017 part of the study.
Lines 405-412: Revisit revised study hypotheses / aims here to clarify the outcomes of the present study
Lines 413-427: Tighten-up this text. There is a lot of supposition and speculation that goes well beyond what the study shows.
Line 416: Where is the evidence that C. fornicata is “stable and persistent”?

Discussion in general needs to be made more succinct and condense speculation

---

## Round 0.2 · Minor Revisions

The revised manuscript has been examined by one of the previous reviewers. Please, address the new minor comments and suggestions. The title requires revision.

Reviewer 1 ·

Basic reporting

The paper is much improved.
I think the authors have engaged with revisions in a positive manner.
Some more speculative claims remain - but have been toned down.

I remain uncomfortable with the title as it is misleading - especially relative to the authors acknowledgement in the rebuttal that they do not have a formal competition test. The narrative that is presented - which is evidenced and supported - is that the decline in oysters is caused by over fishing and environmental change - including pollution; and this left the habitat in a poor soft mud state over time that is low in hard substrate for oysters to re-settle on; slipper limpets then arrived and when at high density could also have negative effects on recovery of oysters but are not indicated in their original decline.

So this is less about competitive exclusion and more about a potential for competition and mixed interactions (predation of larvae is also raised) preventing recovery IF slipper limpets are at higher density. While also acknowledging that they may help when at lower densities by providing some hard substrate.

Therefore could the title not be better phrased - drop the first part and drop competitive exclusion? Change to start with " Is competition with the invasive....."

402-406 - this is unclear due to being broke up with the references. But it is a very important point. The habitat is now generally impoverished - independent of the slipper limpets. The impoverished state is a prime opportunity for expansion of the slipper limpets. And if they reach high densities - "potentially" have competitive interactions with oysters.

Experimental design

From line 282 - The comments on the use of GLM are strange. Its unclear why square root transformations would normalise count data - i.e. the densities. I just plotted the histograms of both the limpet and oyster data and neither are normalized by such transformations? The expected transformations would be log - but please see several papers that discuss why better to use GLM (as below).


All that said - the use of GLM should mean transformations are not required - just use the appropriate family of error distribution to best represent the data. limpet data will conform to neg.bin or poisson. The Oyster data is not in my opinion useable in any standard statistical model format due to the zero inflation.

You also state that your GLM had random effects - which is a different family of models. And you state that replicates were both independent variables and random factors - they should be neither! They are replicates not explanatory variables. And replicates are not different levels within a factor?

I found this all very confusing. Please have this checked and rewrite to make clear.

Validity of the findings

See other points

Additional comments

In the conclusion - given the clear opportunity for competition and other interactions you have identified - perhaps state that the full range of potential +ve and -ve interactions between different stages of different shellfish is as yet unclear and more work needs to be undertaken.

Otherwise thankyou very much for engaging with revisions so positively.

---

## Round 0.3 · Minor Revisions

Please complete the remaining minor revisions as suggested

Reviewer 1 ·

Basic reporting

none further

Experimental design

none further

Validity of the findings

none further

Additional comments

A big thank you to the authors for engaging positively in this review process.

Consider dropping “effective” from title – it flows better without it and focuses on the intervention

Analysis description now much improved – what was the chosen or best fit error structure of the GLM you settled on? This should be stated.

Line 285 – Oyster data “were” – plural?
Line 286-290 – t-tests used while acknowledging Oyster data are highly zero inflated requires caution. The t-test results don't give the degrees of freedom to understand sample size and/or what kind of t-test was undertaken to account for any differences in variation between groups. I suggest the analysis is probably sounds and can keep as is but confirm that same result is obtained with non-parametric equivalent. Only if it is not the same should you consider changing.
Line 286-290 – limpet data compared across sites using t-tests while previous text says harbour*site evaluated using GLM? Is this only to compare between limpets and oysters - if so then this is fine. But see points above about whether any of these parametric approaches are suitable with your oyster data.

I am now really excited by the narrative in the beginning paragraphs of the discussion – it reads well and is supported by good quantitative historical and regional information.

---

## Round 0.4 · accepted · Accept

The authors have satisfactorily addressed all the comments from the reviewers

#